# Study of Root Transparency in Different Postmortem Intervals Using Scanning Electron Microscopy

**DOI:** 10.3390/diagnostics13172808

**Published:** 2023-08-30

**Authors:** Elodie Marchand, Benoit Bertrand, Valéry Hedouin, Xavier Demondion, Anne Becart

**Affiliations:** 1Unité de Taphonomie Médico-Légale et Anatomie, ULR 7367, Faculté de Médecine, Université de Lille, 59000 Lille, Franceanne.becart@univ-lille.fr (A.B.); 2CHRU Nancy, Service de Médecine Légale, 54000 Nancy, France; 3Muséum National d’Histoire Naturelle, Département Homme et Environnement, UMR 7194—HNHP, Institut de Paléontologie Humaine, 75013 Paris, France

**Keywords:** forensic science, taphonomy, sclerotic dentin, root transparency, postmortem changes, scanning electron microscopy, estimated age at death

## Abstract

In the fields of forensics, the identification of human remains is a recurrent problem. The estimated age at death is one of the copious criteria to be evaluated. In adult teeth, the height of the root dentin transparency is used to estimate age. However, in archaeological material, this phenomenon appears inconstant. The aim of this work was to observe the structural modifications of the sclerotic dentin in the teeth for different postmortem intervals. The study included two parts (retrospective and prospective study) with 21 human monoradicular teeth, from bodies donated to medical science with postmortem intervals (PMIs) of 0, 1, 2 and 5 years and archeological excavation. After inclusion based on resin, section and polishing, the samples were analyzed with a scanning electron microscope (SEM) JSM-7800F^®^, and the procedure was completed via a semiquantitative analysis of calcium and phosphorus using EDX microanalysis. The analysis showed the existence of tubular and chemical modifications of sclerotic dentin at different PMIs. Our SEM study allowed us to observe a difference in tubule aspects linked to an increased PMI: the loss of peritubular collar and the lumen obstruction of tubules with a hyperdense material. Microanalysis highlighted variations in phosphocalcic ratios among the different groups, especially in the pulp area and the canine. Our hypotheses that explain these differences are based on the postmortem modifications of the crystals of the mineral phase of sclerotic dentin under the influence of chemical and/or bacterial action.

## 1. Introduction

Dentin is the tooth’s most voluminous tissue [1,2]. It surrounds the pulp and is covered coronally by enamel and radicularly by cementum. It is a tissue composed of 70% mineral matrix (carbonated hydroxyapatite crystals), 20% organic matrix (mainly type I collagen) and 10% water, with thousands of parallel tubules extending from the pulp to the enamel and cementum [3,4]. Dentin is in a perpetual state of remodeling, linked to pulpal vitality, enabling us to distinguish three types of dentin. Primary dentin forms during the development of the tooth until the apexes close. Secondary dentine, with the same composition as primary dentine, gradually replaces it by apposition to the pulp periphery. Finally, tertiary dentine is a reaction tissue produced by odontoblasts in response to physical, chemical or biological aggression [4,5,6]. Like all human tissues, dentin undergoes physiological ageing, starting at the apex and increasing in a coronal direction around the age of 20–25 years; this is known as sclerotic dentin [1,5,7,8]. However, the mechanisms behind this phenomenon are still poorly understood and controversial [9,10]. Some authors state that it is the dissolution of intertubular dentin crystals which precipitate into the tubule lumen [1,11,12], others claim that it corresponds to the centripetal growth of peritubular dentin through continuous mineralization [4,13,14] and, more recently, studies assert that obstruction is linked to odontoblast apoptosis [8,10]. In all cases, this intra-tubular mineralization leads to a partial or even complete occlusion of the tubule [5,15], resulting in a change in the refractive index through the tissue and a translucent appearance in the light [11,16]. Since root transparency is an ante-mortem evolutionary phenomenon, and studies have shown that it correlates well with a subject’s age [7], this parameter is frequently used in techniques for estimating age at death. It is generally associated with other factors, such as periodontal disease [17,18], but can also be used as the sole criterion [19]. The methods described are performed either on whole teeth [18,19,20] or on sectioned teeth [17,19,21]. Although the height of root transparency is commonly used to estimate age at the time of death, particularly in the forensic context of the discovery of altered bodies [9], this phenomenon, observed in subjects who have died recently (several decades), appears to be inconstant in older, particularly archaeological, cases [7]. Associated with the disappearance of the transparency phenomenon, tissue alterations have been described, such as “chalky” dentine [7] or “softened” dentine [22], suggesting that structural modifications of sclerotic dentine occur with postmortem interval. Most studies of sclerotic dentin have focused on the changes observed in relation to secondary dentin. Porter’s electron microscopy and Mandurah’s optical coherence tomography revealed structural differences between the two types of dentin in terms of hydroxyapatite crystal size [1,23]. Raman spectroscopy by Balooch suggested changes in chemical composition, and Kinney found differences in mineralization on X-ray microtomography [11,15]. Postmortem studies have focused on the application of age estimation methods to archaeological materials [7], or on the biological and biochemical phenomena involved in the transformation of a body subjected to different environmental conditions, such as burial [22] or submersion [24]. No basic studies have been carried out on structural, mineral or organic changes in sclerotic dentine after death.

Based on the work carried out on sclerotic dentin and the changes reported by authors, we were interested in the possible changes that could be observed in sclerotic dentin postmortem. Our aim was to propose a methodology for sample preparation and analysis using scanning electron microscopy coupled with microanalysis, complemented by crystallographic analysis using X-ray diffractometry, and then to make observations using repeated measurements to look for variations over time.

## 2. Material and Methods

### 2.1. Population

The study was carried out on twenty-four monoradicular human teeth. In order to be able to make comparisons on the data studied and in particular the chemical variations, we selected the three types of monoradicular teeth (incisor, canine and premolar) for each postmortem interval subgroup. The work was divided into two parts: retrospective and prospective:(1)Retrospective part

This part of the study was carried out on six teeth. Three teeth were donated to science from the Anatomy Laboratory of the Faculty of Medicine of Lille. They were an incisor from a 55-year-old man, a canine from a 60-year-old woman and a lower premolar from a 75-year-old woman. These teeth had been extracted five years before the start of our work. The other three teeth (an incisor, a canine and a premolar) date from the 18th century and come from archaeological excavations.

(2)Prospective part

For this study, eighteen monoradicular human teeth, without morphological, carious or restorative anomalies, were used and separated into 3 groups according to postmortem time: after extraction (group 1), then after conservation for 1 year (group 2) and for 2 years (group 3). We used teeth from body donations to science from the Anatomy Laboratory of the Faculty of Medicine of Lille, free of cancerous, hormonal and/or nutritional pathology. We included teeth from two body donations, a 75-year-old man (subject A) and a 92-year-old woman (subject C). All teeth were extracted by the same odontologist, immediately after the subject’s arrival at the laboratory, with no embalming or freezing procedures applied to the cadavers. In each subgroup, three monoradicular teeth from the same individual were used (one incisor, one canine and one premolar).

### 2.2. Conservation and Preparation

After extraction, each tooth was washed with distilled water and then dried manually. Teeth from groups 2 and 3 were stored individually in hermetically sealed plastic containers under the same conditions as teeth from the retrospective part of the 5-year postmortem period; under a laboratory hood at 20 °C, 60% humidity and an average pressure of 101.7 kPa [25,26]. To consolidate the specimen prior to cutting and polishing, the teeth were embedded in epoxy resin (Araldite^®^ AY 103/Hardener HY 991, Huntsman Advanced Materials, Basel, Switzerland). After curing, each tooth was sectioned longitudinally and then transversely, preserving the sclerotic dentin zone. The cuts were made using a low-speed saw (IsoMet^®^LS, Buehler, IL, USA) and a diamond disc. The final size of each sample was 1 × 1 × 0.5 cm. To obtain a smooth surface, several polishing phases were applied using a hand polisher (1.03.20, Brot technologies^®^, Argenteuil, France). Four successive stages were carried out with silicon carbide polishing discs with grain sizes of P600, P800, P1000 and P1200 (DP 250 AC, Brot technologies^®^, Argenteuil, France) and two successive stages were carried out with polishing discs and alumina suspension, with grain sizes of 2 µm and 0.5 µm. At each stage, a surface inspection was carried out with an optical microscope (LaboVal 4, Zeiss, Oberkochen, Germany) before moving on to the next stage. Finally, the samples were immersed in an ultrasonic bath (Elmasonic S10, Elma ultrasonic, Singen, Germany) of distilled water for 20 min to remove residual particles from the surfaces.

### 2.3. Scanning Electron Microscopy Analysis

As the biological samples were not conductive, a metallization step was carried out. A thin layer of chromium was deposited via sputtering (PECS, Gatan, Pleasanton, CA, USA). After metallization, the samples were analyzed in a scanning electron microscope (JSM-7800F, JEOL, Tokyo, Japan) at 5 kV. Observations were made using secondary electrons, followed by backscattered electrons. For each sample, three areas were selected in the sclerotic dentin as specific locations for photomicrography: the apical zone (just above root closure), in the immediate vicinity of the cementum–dentin junction and in the immediate vicinity of the dentin–pulp interface. In the junction zones, the reference distance was taken at half the height of the root transparency. Magnifications ranged from ×300 to ×20,000. All observations were made by the same observer. As the mineral phase of dentin is composed of carbonated hydroxyapatite crystals [7], we carried out a semi-quantitative analysis of calcium and phosphorus, supplemented by the detection of other elements from the periodic table of elements. These analyses were carried out using energy dispersive X-ray (EDX) microanalysis (AZtec, Oxford Instruments, Abington, UK) with a beam voltage of 10 kV. For each sample, ninety measurements (area 5 µm × 5 µm) were taken in peritubular and intertubular dentin, corresponding to thirty per study area in the locations of the microphotographs.

### 2.4. Statistical Analysis

The results of the microanalyses were entered into a database. We analyzed data from the retrospective and prospective parts of the study separately. For each part, the distribution of the phosphocalcic, sodium and magnesium ratio was analyzed. A multivariate analysis of variance (ANOVA) was performed to analysis the share of variance explained for each factor. For the prospective part of the study, we carried out two correlation tests (Pearson and Spearman) in order to identify a context where the phosphocalcic ratio could best predict the postmortem interval.

### 2.5. Results

(1)Appearance of sclerotic dentin and tubules

SEM analysis for all samples showed a homogeneous appearance of the sclerotic dentin. In secondary and backscattered electrons, structural analysis of sclerotic dentin indicated that tubules were concentrated close to the pulp chamber. However, we observed two types of tubules in sclerotic dentin. Tubules close to the pulp had a rounded appearance, while those close to the cementum–dentin junction were elongated. It was even possible to distinguish the transition zone between the two types (Figure 1).

At higher magnifications (×10,000 to ×20,000) in the areas studied, irregular porous tissue was observed, and at greater depths, a mesh-like structure (Figure 2).

For all samples, regardless of the postmortem interval (PMI), the study of the tubules in the three zones observed revealed similarities in appearance between those close to the pulp and cementum. On the other hand, we observed differences in the apical zone. In fact, for the groups with shorter PMIs, the tubules were surrounded by a denser collar, even for obstructed tubules. This collar was thinner and less visible for teeth with a PMI of 5 years (Figure 3). Tubules in the apical zone formed clusters. The lumen of teeth in groups 1, 2 and 3 was still visible; in contrast, for teeth with a 5-year PMI, the tubules were filled in by a homogeneous, hyperdense material. Finally, at the apex of archaeological teeth, we observed tubules obstructed by hyperdense material, which were grouped together in clusters, but covered by voluminous hyperdense structures (Figure 3).

(2)EDX Analysis

We performed 539 phosphocalcic ratio measurements on the six teeth of the retrospective part and 1620 measurements on the eighteen teeth of the prospective part. Mean measurements were 1.99 (95% CI (1.97; 2), SD 0.15) and 2.04 (95% CI (2.04; 2.05), SD 0.12), respectively. The distribution is shown in Figure 4. A more homogeneous distribution of phosphocalcic ratios is observed in the more recent PMI samples, confirmed via a multifactorial analysis according to tooth type, analysis zone and PMIs.

Concerning the analysis of variance of the phosphocalcic ratio, we observed that in both studies (retrospective and prospective), there was a significant proportion of variance not explained by the selected factors (34.8% and 67.1%), and that for the prospective part, 15.3% of the variance was explained by postmortem interval, followed by tooth type (11.4%) and analysis zone (4.8%). We performed a Pearson–Spearman correlation test between PMIs and tooth type with the analysis area (Table 1). The correlation coefficient was significant, with a negative association, for incisor/pulp (r = −0.35, rs = −0.39), canine/pulp (r = −0.6, rs = −0.71) and canine/cementum (r = −0.57, rs = −0.56).

With regard to the other elements detected, variations were observed only for sodium and magnesium. For sodium concentrations, we took 539 measurements on the six teeth in the retrospective part and 1620 measurements on the eighteen teeth in the prospective part. The mean values were 0.36 (95% CI (0.33; 0.39), SD 0.37) and 0.52 (95% CI (0.5; 0.55), SD 0.49), respectively, with a concentration of 0 in the majority of cases (46.94% and 35.86%). The same number of measurements were taken for magnesium concentrations. Mean values were 0.56 (95% CI (0.52; 0.59), SD 0.42) and 0.79 (95% CI (0.77; 0.81), SD 0.38), respectively. The concentration was 0 in the majority of cases (32.1%) for the retrospective part, but predominantly 0.7 (16.85%) for the prospective part.

## 3. Discussion

Dentin is the main human dental tissue that protects the pulp from damage [11]. The primary organic matrix of dentin, composed of collagen fibers (mainly type I), is secreted by odontoblasts. Under the influence of non-collagenous proteins (phosphoproteins and proteoglycans), dentin becomes progressively mineralized [27]. Crystals are deposited on the collagen mesh in the form of a carbonated apatite plaque. This organo–mineral matrix is crossed by a system of tubules perpendicular to the collagen fibers [12], corresponding to the remaining structure of the odontoblastic processes [28,29]. Once the root has fully formed, secondary dentine is secreted by the pulp. Both types of dentin have the same composition, but are distinguished by a change in the orientation of the tubules [3,30], which we were able to observe under microscopy. Tubules close to the pulp (secondary dentin) were rounded, while tubules close to the cementum (primary dentin) were elongated. We were even able to observe the transition zone between these two types of tubules.

The phenomenon of dentin transparency corresponds to an obstruction of the tubules by a mineral material, which modifies the refractive index of light through the tissue [1,8,11,23]. The origin of this obstruction remains controversial. It may be caused by external processes due to the environment [31] or internal processes originating from the pulp [11,32]. On the basis of these hypotheses, we decided to observe, in sclerotic dentine, the apical zone where the obstruction phenomenon begins, the zone close to the cemento–dentin junction for the influence of environmental factors, and the zone close to the dentine–pulp interface for the influence of pulpal processes. Our results showed that the analysis zone explained only 4.8% of the variance in the phosphocalcic ratio. This factor is therefore not the best one to explain the variations observed. However, in association with tooth type, we could propose a predictive model between phosphocalcic ratio and PMIs. In canine pulp, as the phosphocalcic ratio increased, PMIs decreased, or vice versa, with the best correlation (r = −0.6, rs = −0.71).

The origin of sclerotic dentin is controversial, but the mechanism of its postmortem evolution is even more unclear. Previous studies of archaeological material from different periods have noted differences in the appearance of sclerotic dentine [7,33]. Older dentine was described as “chalky” in the light, but there was no relationship between PMIs and the extent of the changes observed. Very pronounced changes were found in Neolithic and pre-dynastic Egyptian teeth, but were only slightly present in Bronze Age, Mesolithic and 500-year-old teeth [34,35]. Our SEM study showed no difference in tissue appearance between the different PMI groups. It appeared homogeneous, rather porous, and organized in mesh. However, the appearance of the tubules in the apical zone was different. The peritubular collar, presenting a “coffee-bean” appearance to the teeth observed after avulsion, was less prominent as PMIs increased. As peritubular dentin is merely mineralized tissue [36], its “disappearance” should correlate with the decrease in the phosphocalcic ratio. This is what we found. Indeed, chemical analysis showed variations in the phosphocalcic ratio according to PMIs (2.04 for recent teeth and 1.99 for old teeth). We can propose several hypotheses to explain the differences in phosphocalcic ratios observed according to PMIs. This could be due to the dissolution of peritubular dentin crystals and precipitation in the tubule lumen. Indeed, we observed that the tubule lumen of teeth with PMIs of 2 years and 5 years appeared to be obstructed by hyperdense material, which could be crystalline. This hypothesis is close to Porter’s concerning the formation of sclerotic dentin from the dissolution of intertubular dentin crystals [1]. We were able to confirm the presence of phosphorus and calcium in the tubular lumen, but were unable to quantify the phosphocalcic ratio due to irregularities in depth. Porter has suggested the involvement of chemical species such as fluorine to explain this phenomenon [1]. Although we observed no variation in the fluoride content of our samples, we did observe variations in two other elements: sodium and magnesium. These two elements could be involved in variations in the phosphocalcic ratio. Other studies have shown substitutions by magnesium, attributed to exchanges during the dissolution and recrystallization phases of hydroxyapatite crystals in vitalized [37] or fossilized [38] teeth. This dissolution of peritubular dentin crystals could also be due to the production of acid ions by bacteria [39,40]. In our study, the bacteria could have originated in the oral cavity, as the extracted teeth had not been sterilized. Bacterial action could also cause the destruction of the collagenous structure of dentin tissue, which could influence internal controls and tissue acidity, increasing the availability of ions present in large quantities, such as calcium and phosphorus [10]. These phenomena would occur over short postmortem periods (less than a hundred years) and the phosphocalcic ratio would then remain stable. This is what Beeley and Lunt found in their study of teeth from different periods (Neolithic, Bronze Age, 9th–11th centuries and 15th–17th centuries), where no significant differences in calcium and phosphorus ratios were found between the different groups [22]. Finally, Patonaï et al. have demonstrated a link between crystallinity and demineralization in forensic and archaeological bones [41]. They showed that as PMI increases, the phosphocalcic ratio decreases, but the crystallinity index increases. Crystals appear larger and more ordered. An X-ray diffractometry analysis is currently underway to study variations in crystallinity as a function of PMIs in our different groups.

We chose SEM for this study because it enables us to visualize the structure of a sample without destroying it completely. However, the quality of the analysis depends on the quality of the sample preparation. To this end, we added a polishing step using aluminum suspensions. We decided to work with a voltage of between 5 and 15 keV to limit the risk of charging effects on our samples, which could be damaged by prolonged exposure. In addition, working at low voltage enabled us to obtain more precise information on tissue topography. An analysis combining secondary and backscattered electrons should be systematically carried out. In fact, backscattered electron analysis enabled us to observe topographical contrasts and identify obstructed tubules invisible to secondary electrons, corresponding to a contrast in density and/or composition. Our results are limited by a small sample size, reflecting two difficulties. The first concerns the exclusion criterion for dental pathologies. Indeed, we decided to study sclerotic dentin, and therefore the physiological ageing of secondary dentin; the teeth therefore had to be free of pathologies, particularly carious pathologies. People who donate their tissue for scientific studies are more likely to be elderly subjects who often neglect dental care as they age, or who have no teeth left. The second difficulty was obtaining materials. In France, donations from scientific organizations remain limited and do not allow us to obtain a large number of samples for study. Finally, given the small number of samples and, above all, the small number of different subjects, we were unable to study the influence of age or sex on the changes observed. However, multifactorial analyses of the phosphocalcic ratios, particularly for the prospective part, showed that the subject factor explained only 1.4% of the variance, compared with 15.3% for the postmortem interval. This suggests that sex and age play only a minor role in chemical variations. Premortem studies of sclerotic dentin found gender differences in appearance and growth, but no significant differences. Age has an influence on sclerotic dentin growth, but Mandojana observed no structural differences in postmortem study [42]. However, we did manage to collect an identical number of incisors, canines and premolars from two subjects of different sexes and ages, enabling us to carry out an inter- and intra-individual comparative study. Finally, this work focused on very short PMIs, which could be of real interest in forensic science.

## 4. Conclusions

Our SEM study enabled us to observe the existence of tubular modifications in sclerotic dentin with different PMIs. By complementing this study with chemical microanalysis, we observed the presence of variations in the phosphocalcic ratio with PMIs and the possibility of proposing a predictive model. Finally, we have also highlighted variations in other elements that could be involved in these modifications. Further work is required to understand the origins of these changes. These observations, carried out on a small sample of analyses, should be carried out on a larger scale with longer postmortem times. They should focus on the canine, which showed a significant variation in phosphocalcium ratio between the PMIs studied.

## Figures and Tables

**Figure 1 diagnostics-13-02808-f001:**
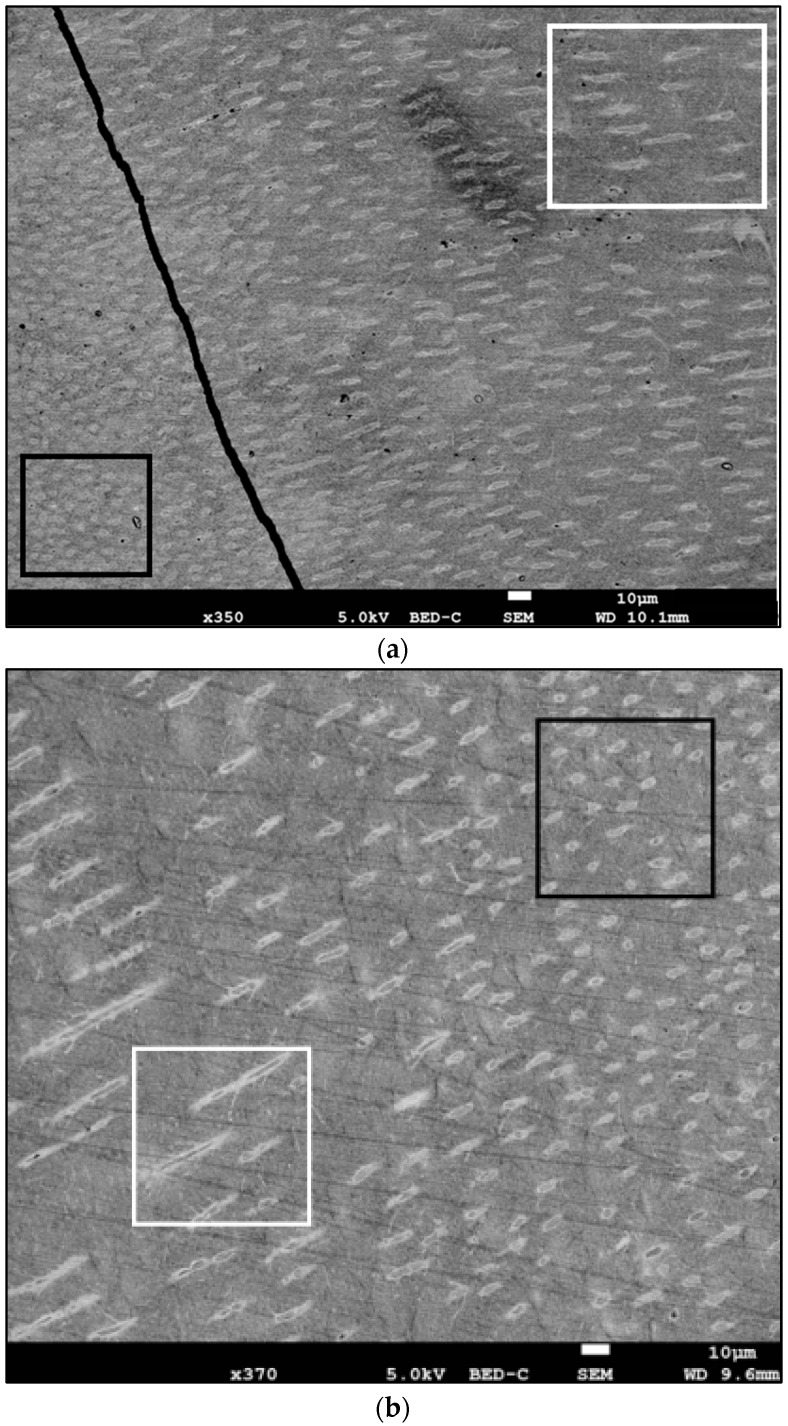
Photomicrograph obtained via SEM in backscattered electrons (magnification ×190 and ×350–370)—(**a**) human tooth with a PMI of 0 y, (**b**) human tooth with a PMI of 1 y and (**c**) human tooth with a PMI of 2 y. Tubules near pulp chamber rounded (black square), tubules near cemento–dentinal junction lengthened (white square).

**Figure 2 diagnostics-13-02808-f002:**
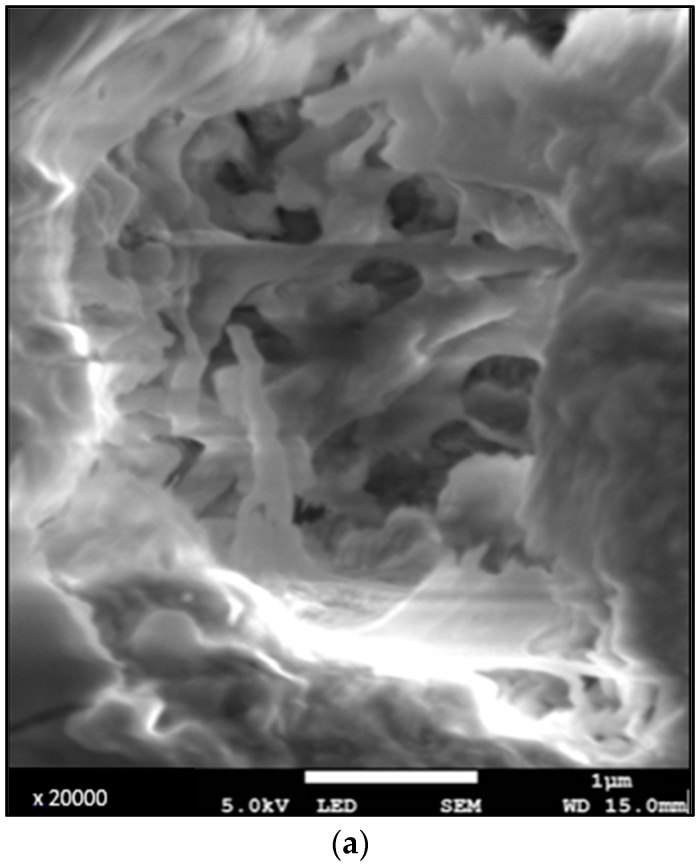
Photomicrograph obtained via SEM in secondary electrons (magnification ×15,000 and ×20,000)—(**a**) human tooth with a PMI of 5 y, (**b**) human tooth with an archeological PMI. Structure in mesh in lumen of tubule.

**Figure 3 diagnostics-13-02808-f003:**
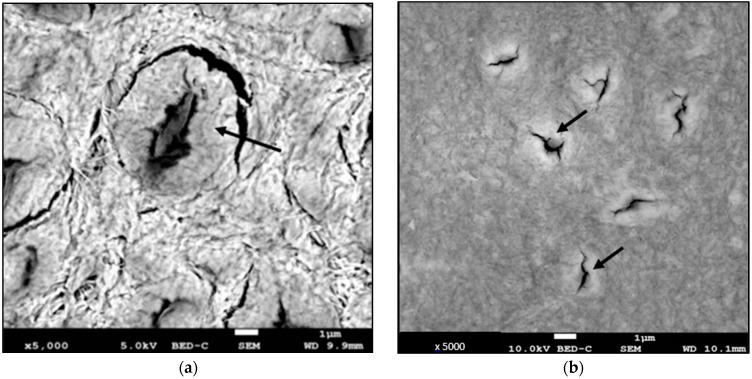
Photomicrographs obtained via SEM (magnification ×4000 and ×5000). Apical area: Tubules surrounded by a denser collar (black arrows). (**a**) Human tooth with a PMI of 0 y, (**b**) human tooth with a PMI of 1 y, (**c**) human tooth with a PMI of 2 y, (**d**) human tooth with a PMI of 5 y, (**e**) human tooth with an archeological PMI. Tubules obstructed by a hyperdense material (white arrow), covered by voluminous hyperdense structures.

**Figure 4 diagnostics-13-02808-f004:**
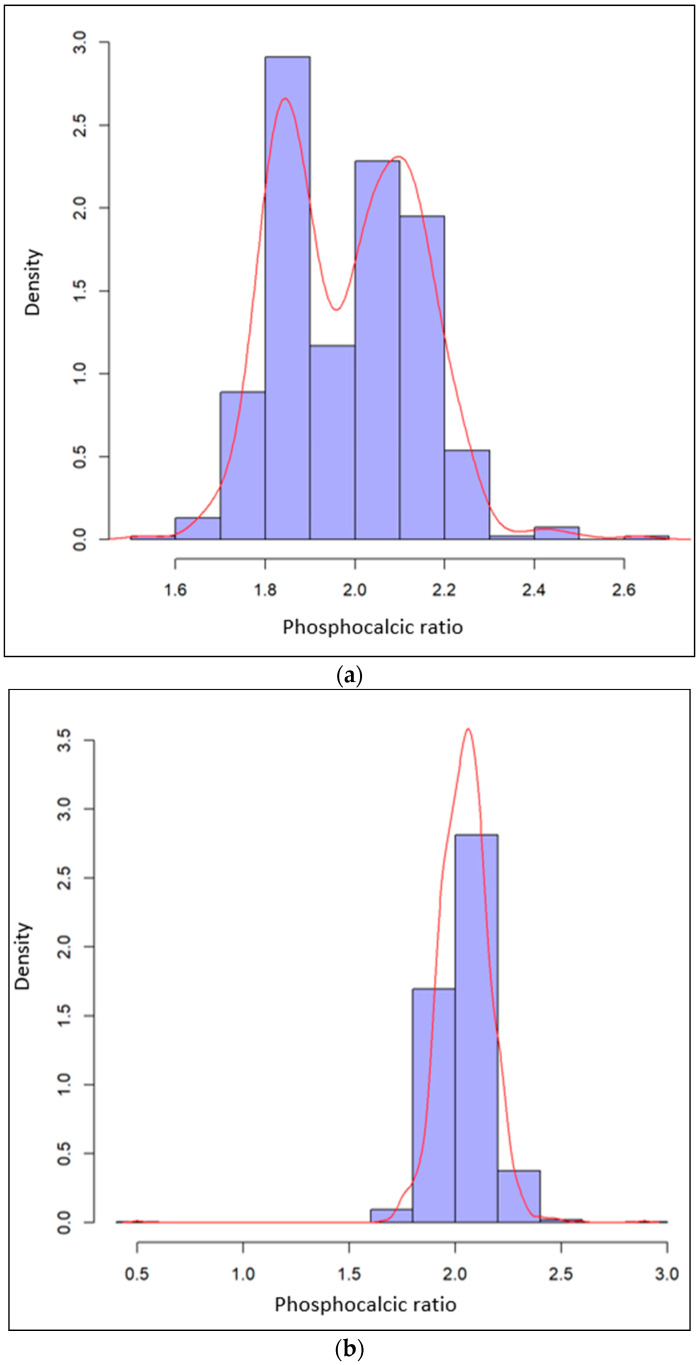
(**a**) Distribution of the 539 phosphocalcic ratio measurements in the samples of the retrospective part and (**b**) distribution of the 1620 phosphocalcic ratio measurements in the samples of the prospective part.

**Table 1 diagnostics-13-02808-t001:** Pearson and Spearman association test for samples from the prospective part of the study between type of tooth/analysis area and postmortem interval.

Type of Teeth	Area of Analysis	r	r^2^	*p*	rs	rs^2^	*p*
Incisor	Pulp	−0.35	0.12	1.53 × 10^−6^	−0.39	0.16	2.81 × 10^−8^
	Cement	0.03	0.001	0.64	−0.01	1 × 10^−4^	0.87
	Apex	−0.02	2.6 × 10^−4^	0.83	−0.03	8 × 10^−4^	0.70
Canine	Pulp	−0.69	0.48	4.99 × 10^−27^	−0.71	0.50	9.30 × 10^−29^
	Cement	−0.57	0.32	1.09 × 10^−16^	−0.56	0.31	2.71 × 10^−16^
	Apex	0.02	3.5 × 10^−4^	0.80	−0.02	6 × 10^−4^	0.74
Premolar	Pulp	−0.03	8.6 × 10^−4^	0.69	0.03	0.001	0.65
	Cement	0.02	4.1 × 10^−4^	0.79	0.03	8 × 10^−4^	0.71
	Apex	0.06	0.004	0.39	0.14	0.02	0.07

r: Pearson’s linear correlation coefficient. rs: Spearman’s correlation coefficient. r^2^ and rs^2^: Share of explained variance. *p* significant < 0.05.

## Data Availability

The data is available from the Unité de Taphonomie Médico-Légale et d’Anatomie de Lille.

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
