# Peer review of "Study of Root Transparency in Different Postmortem Intervals Using Scanning Electron Microscopy"

_diagnostics, 2023, doi:10.3390/diagnostics13172808_

Round 1
Reviewer 1 Report
Q1:The manuscript structure and format have to be checked, as there are several errors such as a mix of fonts in lines 47 to 76, and figure 4 lacks X-axis and Y-axis labels.
Q2:The article is titled “Modifications in Root Transparency with Different Postmortem Intervals: Study with Scanning Electron Microscopy”. But the manuscript mainly focuses on the characterization of SEM microstructures and the EDX analysis of phosphocalcic ratio, the detailed explanation of the relationship between structural changes, phosphocalcic ratio, and root transparency is not provided.
Q3:In the part “Appearance of sclerotic dentin and tubules”, figure 2 should be presented with priority, and it should also be supplemented with different PMI for comparison. And then the corresponding high-magnification microstructures of the tubules should be provided, such as figure 1. Furthermore, the high-magnification SEM images are not sufficiently clear. Please consider the rationality of the metallization step, which could potentially impact magnification imaging.
Q4:For figure 3 a to d, a uniform magnification scale should be applied. The source of figure 3e needs to be explicitly stated, and the white arrow are not visible in the image.
Author Response
Response 1 :
The manuscript have checked and the style has been standardized.
In figure 4 X-axis and Y-axis have been added
Response 2 :
The title of the article has been changed to "Study with Scanning Electron Microscopy of Root Transparency in Different Postmortem Intervals".
Indeed, with our current results, on our sampling it was not possible to establish a relationship between the tubular structural modifications observed and the variations in the phosphocalcic ratio.
Response 3 :
Figure 2 became Figure 1 (and vice versa) and was placed first in the part "Appearance of sclerotic dentin and tubules" section. Two more images have been added for post mortem times of 0 years and 5 years
In Figure 1 (now 2) we have added a high magnification image of the mesh structure in the lumen of a tubule of an archaeological tooth. Unfortunately, we do not have any other images for the other postmortem intervals because exposure to this magnification with a high voltage tended to destroy the tissue, so we preferred to avoid this overexposure for taking photographs.
Response 4 :
In figure 3 we have modified the image of the tubules so that the magnification corresponds to x5000 for the teeth of PMI 1 year, but we do not have this magnification for a PMI 2 years. For figure 3 (e) we have specified that it is an archaeological individual and we indicated the tubules by a white arrow.
Reviewer 2 Report
Dear authors,
congratulations for the interesting topic chosen. It is possible to detail the following aspects:
- how was the number of teeth taken in the study calculated?
- do you consider that the general pathology of the patient can influence the presence of sclerotic dentin?
- can the changes in dentine be correlated with the age of the patient at the time of death?
Author Response
Thank you for your proofreading and your questions:
- The number of teeth used unfortunately could not be calculated, it was dependent on donations of bodies to science in the Faculty of Medicine which is very limited. On the other hand, in order to have a minimum comparison, we chose to obtain the same number of types of teeth for each interval studied (one incisor, one canine and one premolar). The following paragraph has been added in the manuscript at the beginning of the material and method part "In order to be able to make comparisons on the data studied and in particular the chemical variations, we selected the three types of monoradicular teeth (incisor, canine and premolar) for each postmortem"
- Concerning the general pathologies, it is possible that certain metabolic pathologies can cause chemical modifications in the sclerotic dentin, since this one probably undergoes influences coming from the environment but also from the pulp and therefore from the patient himself. Our study is extremely preliminary and does not allow us to provide an answer in this sense, but could be one of the indicators of variation in phosphocalcic ratios not explained by the post-mortem period, the type of teeth or the area studied, just like the subject's lifestyle.
- Regarding the influence of sex and age, we were not able to really study it in relation to our sampling. We added at the end of the discussion in our manuscript the following part to provide an answer to this question "Finally, given the small number of samples and, above all, the small number of different subjects, we were unable to study the influence of age or sex on the changes observed. However, multifactorial analyzes of the phosphocalcic ratios, particularly for the prospective part, showed that the subject factor explained only 1.4% of the variance, compared with 15.3% for the post-mortem interval. This suggests that sex and age play only a minor role in chemical variations. Pre-mortem studies of sclerotic dentin found gender differences in appearance and growth, but no significant differences. Age has an influence on sclerotic dentin growth, but Mandojana observed no structural differences in postmortem study. " (reference 43)